# Impact of High Temperature on Post-Exercise Albuminuria in Dogs

**DOI:** 10.3390/ani10111988

**Published:** 2020-10-29

**Authors:** Urszula Pasławska, Barbara Szczepankiewicz, Aleksandra Bednarska, Robert Pasławski

**Affiliations:** 1Institute of Veterinary Medicine, Faculty of Biological and Veterinary Sciences, Nicolaus Copernicus University in Torun, ul. Gagarina 7, 87-100 Torun, Poland; urszula.paslawska@umk.pl (U.P.); r.paslawski@umk.pl (R.P.); 2Department of Internal Medicine and Clinic of Diseases of Horses, Dogs and Cats, Faculty of Veterinary Medicine, Wroclaw University of Environmental and Life Sciences, Pl. Grunwaldzki 47, 50-375 Wroclaw, Poland; aleksandra.bednarska@upwr.edu.pl

**Keywords:** canine, sport, diagnostic markers, kidney, proteinuria, urinalysis

## Abstract

**Simple Summary:**

The literature suggests that strenuous exercise and exposure to high temperatures may cause physiologic proteinuria, but there are no available data showing the effect of high temperature on the occurrence of post-exercise albuminuria in dogs. The goal of the present study was to assess impact of high temperature on the occurrence of albuminuria. A period of about 30–40 min of intensive exercise at high temperatures did not lead to increased albuminuria. This suggests that there is no need to limit physical activity before urine tests in dogs, even at high temperatures, before a urinalysis.

**Abstract:**

The literature suggests that strenuous exercise and exposure to high temperatures may cause physiologic proteinuria, but to our knowledge there have been no studies that have assessed the effect of high temperatures on the occurrence of post-exercise albuminuria in dogs. The goal of the study was to assess the impact of high temperatures on the occurrence of albuminuria. Thirteen healthy adult dogs—eight female (62%) and five male (38%) had to run 5 km at a temperature of 25 °C in grasslands which took about 30–40 min. Dogs underwent clinical examinations: echocardiography, abdominal ultrasound, blood hematology and biochemistry and urinalysis, including the ratio of albumin to creatinine (UAC). Baseline UAC was on the borderline of statistical significance for female dogs, but not for male dogs, before and after exercise. UAC was 0.31 ± 0.56 mg/mmol for female dogs and 0.36 ± 0.60 mg/mmol in male dogs before exercise. Immediately after exercise, UAC was 0.51 ± 0.58 mg/mmol in female dogs and 0.31 ± 0.40 mg/mmol in male dogs. Thus, a period of about 30–40 min of intensive exercise at high temperatures (25 °C) did not lead to increased albuminuria. This suggests that there is no need to limit the movement activity before urine tests in dogs, even at high temperatures, before urinalysis.

## 1. Introduction

Urinalysis (UA) was the first laboratory test performed in medicine and has been used for several thousand years [1]. One of the most important parameters evaluated in UA is the detection of protein in urine, including albumins, globulins and Bence Jones proteins. Proteinuria of renal origin results from two major mechanisms: the first is a loss of selective glomerular filtration as a result of podocyte loss (with a negatively charged sialoglycoprotein coat that contributes to the charge selectivity of the filtration barrier) leading to increased amounts of plasma protein in the filtrate (urine). The second is an impaired resorption of the filtered protein [2,3]. Albumin is the predominant protein in canine and feline urine in both healthy and renal disease subjects [4]. The urine of healthy dogs contains <1 mg/dL albumin [4].

According to available reports, physiologic proteinuria may be caused by strenuous exercise, seizures, fever and exposure to extreme temperatures (heat or cold) [5,6]. Previous studies on dogs demonstrated that microalbuminuria developed in 0 to 15% of animals after exercise, while in 15% of dogs, positive microalbuminuria was present both before and after exercise [7]. These results differed from data obtained from studies on humans, which showed that transient albuminuria is common after intense exercise, especially at high temperatures, with prevalence ranging from 18% to 100%, depending on the type and intensity of exercise and temperature during exercise [5,7]. Both exertion and heat stress result in a decrease in glomerular filtration and renal blood flow (notwithstanding an increase in mean arterial pressure), which reduces urine output [8,9]. The reduction in effective renal flow to 25% leads to post-exercise albuminuria (PA). Currently, veterinarians suggest to reduce physical effort a day before urine examination. Limitation of physical activity before urinary examination is very impractical due to the fact that many dogs are physically active daily. 

The main purpose of this research was to determine whether exercise in high outside temperatures would lead to the development of albuminuria in dogs.

## 2. Material and Methods

### 2.1. Animals

This study included 8 ovariohysterectomized female dogs, 3 castrated dogs, and 2 intact male dogs that were presented for prophylactic examinations at the Department of Internal Diseases with the Clinic of Horses, Dogs and Cats, Faculty of Veterinary Medicine at the University of Life Sciences in Wroclaw. Represented breeds included: 3 Nova Scotia Duck Tolling Retrievers, 3 mixed breeds, 2 Golden Retrievers and one of each Flat Coated Retriever, German Shepherd, Belgian Shepherd Malinois, Border Collie and Weimaraner.

### 2.2. Study Protocol

The inclusion criteria for dogs were: medium and large dogs. The dogs were characterized by high physical activity. Health status was proved based on the anamnesis, clinical examination, blood tests, ultrasonography and echocardiography examinations.

The exclusion criteria were: pregnancy, lactation, periods of growth and convalescence, abnormalities in clinical, blood, urine, ultrasound and echocardiography examination and the presence of orthopaedic disease.

Preliminary examination (except urine examination) was performed one day before exercise, allowing for selection of healthy dogs.

### 2.3. Clinical Examination 

The standard clinical examination included: examination of the mucous membranes, lymph nodes, auscultation of the heart and abdomen palpation and temperature measurement. Body temperature was measured with rectal thermometers.

### 2.4. Haematological and Biochemical Blood Test

Blood sampling was preceded by a minimum 8 h fasting period, during which dogs had free access to fresh water. Blood samples were collected in the morning hours from the cephalic vein into EDTA blood collection tubes. Hematological examination was performed using a Horiba ABC animal blood counter (Horiba ABX SAS, Montpellier, France) and an IDEXX X LaserCyte (IDEXX, Westbrook, Maine, USA) haematology analyser. The concentrations of haemoglobin (HB), Ht, RBCs, WBCs were measured. Blood samples for biochemical analysis were collected from the same vein into plain tubes and left for 15 min at room temperature. Next, the samples were centrifuged at 2000× *g* for 10 min. Urea and creatinine were assessed in serum using the TermoScientific Konelab Prime 30ISE (Thermo Fisher Scientific, Vantaa, Finland) biochemical analyser. Blood tests were performed to select dogs that had all parameters within the reference range.

### 2.5. Urinalysis (UA)

Samples of the first morning urine, and the first urine after exercise (about 15 min after exercise) were collected by the owner into a sterile container during spontaneous urination. Dogs did not have access to water during exercise. The UA was carried out immediately after receiving the urine samples. A physico-chemical examination of the urine included analyses of its color, transparency, urine specific gravity (uSG), pH, protein, albumin, creatinine, glucose, blood, acetone, bilirubin, ketones and urobilinogen levels. The uSG was analysed using a clinical handheld refractometer that was calibrated with distilled water prior to the beginning of each test and reviewed periodically between sample measurements, with the same individual measuring all urine samples. Additionally, microscopic examination was performed to examine the urine sediment. Urine albumin, protein and creatinine were measured in urine supernatant using the TermoScientific Konelab Prime 30ISE (Finland) biochemical analyser in the analytical laboratory. Urine albumin was measured using calibration kit-SpeciCal and control- SpeciTrol and SpeciTrol High. The urine albumin to creatinine ratio (UAC) was calculated by dividing the values of albumin concentration by the values of creatinine concentration in urine. Albuminuria may be considered physiological (normoalbuminuria) if the value of UAC is smaller than 30 mg/mmol. Microalbuminuria is observed when the values of UAC are within the 30–300 mg/mmol range, and macroalbuminuria is diagnosed when the UAC values are above 300 mg/mmol [10].

### 2.6. Ultrasonographic Evaluation

A standard abdominal ultrasound examination was carried out using the Hitachi Aloka F37 (Hitachi Aloka Medical, Ltd., Tokyo, Japan) machine with a 5–10 MHz microconvex and linear probe. Ultrasound examination allowed us to exclude dogs with concurrent neoplastic pathologies. Additionally, careful examination of urinary bladder and kidneys was performed to exclude pathological changes in the urinary system.

### 2.7. Echocardiographic Evaluation

Transthoracic echocardiography was performed using a Hitachi Aloka F37 (Hitachi Aloka Medical, Ltd., Tokyo, Japan) echocardiograph with a 5.0–7.5 MHz sector probe. The aorta diameter (Ao), left atrium size (LA), left atrium size/aorta diameter (LA/Ao) were performed using B-Mode in the right parasternal short axis view. M-mode right parasternal long axis view was performed to calculate the end-diastolic (LVIDd) internal diameter of the left ventricle, shortening fraction (FS) of the left ventricle, ejection fraction (EF), using the Teicholz formula. Echocardiography was based on previously published values [11]. Echocardiography examination was performed to exclude heart disease, which could cause albuminuria.

### 2.8. Exercise

Exercise was performed at the end of June and started when the outside temperature reached 25 °C. In this study, 25 °C was regarded as “hot” based on typical temperatures of the geographic localization of the study. The average annual temperature in Poland is about 6–8 °C, thus 25 °C is considered to be a high temperature. All dogs trained at the same time and ran 5 km on grassland at a steady pace. The duration of intense exercise was 30–40 min. The dogs did not have access to water during the exercise. During and after the exercise, stress appeared to be minimal in all dogs, as judged subjectively by the examiner based on dogs’ outward behaviour. Most dogs panted for several minutes after exercise but rapidly returned to the same respiratory patterns and behaviours shown before exercise. All dogs drank water after the exercise and urinated approximately 15 min after exercise. No dog showed signs of heat stress that required medical intervention.

### 2.9. Ethics Approval and Consent to Participate

This study was performed according to the national and institutional guidelines on the use of animals in clinical research according to the Polish legal act concerning experiments performed on animals of 21 January 2005 (Dz. U. z 2005 r. Nr 33, poz. 289 z późn.zm). A routine veterinary examination (blood and urine sampling, cardiac examination and ultrasound) was performed in accordance with the above mentioned legal act, and a written ethical approval from the Local Ethical Committee before the beginning of the study was not necessary. The training was conducted based on the knowledge of an experienced dog trainer.

### 2.10. Statistical Analysis

Data editing and statistical analyses were performed using StatSoft Statistica PL 12.0 software. The normal distribution of data was expressed as median and standard deviation (±SD). Data were analysed using the *t*-Student’s test. Simple regression tests was used to describe the relationships between variables. A *p* value < 0.05 was considered statistically significant. Due to the limited number of observations, a maximum of two covariates were included in the model. 

## 3. Results

Median values of clinical results of all dogs subjected to intensive physical activity at high temperatures (25 °C) are shown in Table 1 and the results of body temperature and UA before and after exercise for all dogs, females and males are presented in Table 2. 

Body temperature was insignificantly higher in all dogs after exercise in comparison to the values registered before exercise. uSG was insignificantly lower in all dogs, female and male dogs, after exercise than before exercise. We observed a wide variability in uSG values after exercise. uSG ≤ 1.020 was found in 62.5% of female dogs and 60% of male dogs. Urine albumin was significantly higher when all dogs were grouped together; however the increase was not significant in female dogs, whereas male dogs showed significantly lower values of urine albumin after exercise than before. Urine creatinine was lower in all dogs grouped together; however when male dogs were analysed separately the result was on the border of significance, whereas female dogs showed statistically insignificant decreases in creatinine concentration after exercise than before. UAC was significantly higher in all dogs grouped together and in female dogs after exercise, but in the group of male dogs the value was slightly and insignificantly lower after exercise. After exercise in female dogs UAC changed a maximum of 2.12 fold (mean 1.39 ± 0.87), while in male dogs UAC changed a maximum of 4.33 fold (mean 1.79 ± 1.81). When the results of UAC before exercise were analysed together for all dogs included in the study the values changed max. 4.33 fold (mean 1.51 ± 0.67).

## 4. Discussion

Our research showed that healthy dogs subjected to 30–40 min physical activity, even at high outdoor temperatures, do not develop albuminuria. Results obtained in the present study, showing comparable values of UAC before and after exercise in training dogs, are contradictory to the data obtained from research on human subjects, which demonstrated that in order to obtain a reliable urinalysis result, people should not do strenuous exercises 24 h before the test, especially at high temperatures. 

Until now, the veterinarian had to repeat UA of dogs trained at high temperatures, because it was suspected that such conditions may lead to increased albuminuria. Albuminuria was reported in dogs with underlying medical conditions such as diabetes mellitus, hypertension, after splenectomy, in dogs with heartworm, as well as with renal disease and heart disease [12,13,14,15,16]. Albuminuria is indicative of active, progressive renal injury, and should prompt further investigation to detect any infectious, inflammatory or neoplastic diseases that might be the underlying cause of renal disease [17]. Albuminuria detected before exercise causes increased presence of albumins in urine after exercise (PA) [7].

According to Gary et al. [7], in all dogs that did not have albuminuria before physical exertion no PA was found, while in 15% of cases positive microalbuminuria was present both before and after exercise. Dogs with single albuminuria results in urine should repeat urinalysis twice within 3–6 months, and perform biochemical blood examination (including urea creatinine, symmetrical dimethylarginine (SDMA), C-reactive protein (CRP), as well as glucose and fructosamine if needed), as well as ultrasound examination of the abdomen with particular emphasis on the urinary tract (kidneys, ureters and urinary bladder and urethra) to exclude pre-renal, renal and post-renal causes of albuminuria. In addition, neoplastic diseases should be excluded by using abdominal ultrasound examinations and chest radiography. Congenital and acquired heart disease and systemic hypertension should also be excluded by echocardiography and Doppler or oscillometric measurements [18].

The term albuminuria is now preferred in nephrology as a substitute for macroalbuminuria [17]. Albuminuria is detectable by a routine dipstick urine protein measurement in contrast to normoalbuminuria and microalbuminuria [17]. Additionally, it is worth noting that albuminuria is a better marker of kidney disease in dogs than in cats, probably owing to the pathophysiology of chronic kidney disease (CKD) in cats, which is characterized by low-grade proteinuria and less glomerular involvement than in dogs [19,20]. 

In the present study conducted on healthy dogs, mean UAC values after exercise were slightly increased, but still remained within the physiological range, whereas in some dogs UAC was even decreased after exercise, suggesting that no PA was detected in dogs.

Morning urine is always more concentrated that urine obtained during the day, which was also confirmed in our research, but the differences were not statistically significant. As written, the study appears to have been carried out on two different groups of dogs and comparison of the urine samples collected during the day in training and non-training dogs revealed decreased urine creatinine concentrations in the exercise groups, indicating that either less creatinine was excreted into urine or the urine samples were more diluted [21,22]. Therefore, it appears likely that creatinine was filtered from the serum into the urine in a different way after training at high temperatures than without exercise [23,24,25]. uSG in the first morning urine provides an easy way to determine hydration status and the response to dehydration. uSG for healthy dogs should be within a range of 1.011 to 1.060 [26], which means that all dogs included in this study had uSG within the reference values. Based on standards used for athletes in medicine the state of hydration after exercise was classified using urine specific gravity values, as follows: uSG ≤ 1.020 good body hydration, uSG > 1.020 state of dehydration [27]. However, assessing hydration status in dogs after exercise at high temperatures requires more research. Undoubtedly, physical exercise has an influence on the neurohormonal balance, body hydration, urine concentration and body temperature. Temperature in dogs is mainly regulated by respiratory exchange and associated with evaporative heat loss [28,29]. The combination of high external temperatures, exercise and heat production by muscle mass exceeds the evaporative cooling capacity through panting, and leads to elevated body temperature. 

The limitation of our study was the lack of serum albumin results. This does not affect the results of urine albumin in healthy dogs, however in dogs with urinary or gastrointestinal tract diseases, inflammation or neoplasia, serum albumin could affect the result of urine albumin concentration. A limiting factor of our research was the small group of dogs tested. In addition, we examined only regularly exercising medium and large dogs. We have not examined dogs of small and giant breeds, non-exercising dogs in poor or medium physical condition, or suffering from any illnesses. Additionally, dogs were subjected to physical exercise at distance of 5 km, and the effect of PA on longer distances has not been investigated to avoid violating animal welfare and overheating the training dogs.

## 5. Conclusions

In healthy dogs, about 30–40 min of intensive exercise at high temperatures (25 °C) does not cause albuminuria. In dogs, mean UAC values may slightly increase after exercise, but still remain within the physiological range, while in some dogs UAC may even decrease after exercise. This suggests that there is no need to limit the physical activity in dogs before urinalysis.

## Figures and Tables

**Table 1 animals-10-01988-t001:** Median values of clinical results of 13 dogs subjected to intensive physical activity at high temperatures (25 °C).

Variable	Median	Standard Deviation
Clinical data		
Age (years)	4.5	2.1
Weight (kg)	27.1	6.5
Haematological results		
RBCs (T/L)	6.9	1.2
HB (mmol/L)	10.0	1.3
Ht (%)	49.6	5.7
WBCs (G/L)	7.9	2.7
Biochemical results		
Serum urea (mmol/L)	6.3	2.0
Serum creatinine (µmol/L)	116.5	17.6
Echocardiographic results		
LA/Ao	1.3	0.1
LVIDd (mm)	31.0	6.7
FS (%)	37.0	13.5
EF (%)	59.9	15.6
HR (min^–1^)	124.0	18.0

EF—ejection fraction, FS—shortening fraction, HB—haemoglobin, LA/Ao—left atrium size/aorta diameter, LVIDd—end-diastolic internal diameter of the left ventricle.

**Table 2 animals-10-01988-t002:** Body temperature and urinalysis before and after exercise in dogs.

Variable	Dogs	Before Exercise	After Exercise	*p*-Value
Body temperature (°C)	All dogs	38.22 ± 0.45	39.68 ± 0.57	NS
	Female dogs	38.32 ± 0.43	39.65 ± 0.73	NS
	Male dogs	38.24 ± 0.29	39.74 ± 0.21	NS
uSG (mg/L)	All dogs	1.037 ± 0.011	1.029 ± 0.021	NS
	Female dogs	1.041 ± 0.012	1.023 ± 0.014	NS
	Male dogs	1.030 ± 0.008	1.037 ± 0.030	NS
pH	All dogs	6.7 ± 0.5	7.1 ± 0.7	NS
	Female dogs	6.6 ± 0.3	7.2 ± 0.8	NS
	Male dogs	7.0 ± 0.7	6.9 ± 0.6	NS
Urine albumin (mg/L)	All dogs	5.4 ± 7.8	5.7 ± 8.0	0.01
	Female dogs	5.3 ± 8.6	7.5 ± 9.7	NS
	Male dogs	7.5 ± 9.7	2.8 ± 3.1	0.03
Urine creatinine (mmol/L)	All dogs	22.3 ± 9.2	14.4 ± 10.5	0.01
	Female dogs	20.3 ± 4.1	13.4 ± 9.6	0.018
	Male dogs	25.6 ± 14.3	15.9 ± 12.8	NS
UAC (mg/mmol)	All dogs	0.33 ± 0.56	0.43 ± 0.47	0.0004
	Female dogs	0.31 ± 0.56	0.51 ± 0.58	0.049
	Male dogs	0.36 ± 0.63	0.31 ± 0.40	NS

UAC—urine albumin to creatinine ratio, uSG—urine specific gravity, NS—not statistically significant.

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
