# Peer review of "Impact of High Temperature on Post-Exercise Albuminuria in Dogs"

_animals, 2020, doi:10.3390/ani10111988_

Round 1

Reviewer 1 Report

Thank you for your efforts in clarifying and improving your manuscript

Reviewer 2 Report

The article has improved substantially, however there are a number of points to improve.

Reference values ​​for urine albumin and UAC are not mentioned in materials and methods. Therefore, it is difficult to know whether or not the results are within the reference values, so it is not possible to test the statement on line 204.

Lines 116 - 117: The sample is said twice to have been measured in the supernatant.

In terms of results, the study considers that a P value less than 0.05 is considered statistically significant. However, in lines 192-193 it is said that the results in the females were on the edge of significance, when according to table 2, the P value was 0.018.

In the discussion the different results obtained between males and females are not addressed.

Lines 216 - 217. It is not understood what PA means. I assume it will be "presence of albumin", but it is not clear.

Line 218: Reference to Gary et al. should be immediately after “et al”

Line 221: Instead of “repeated”, I think it is “repeat”.

Lines 218 - 220: here it is said that in dogs that did not have albuminuria before physical exercise it was also not found after, but in the introduction (lines 152-154), it is said that 0 to 15% of the animals developed microalbuminuria after exercise. Please, clarify

Lines 232 - 236: these are reference values ​​that must therefore be specified in the methodology.

Lines 236-237: is it a result? In that case, it must go in its corresponding section.

Lines 245-247: As written, the study appears to have been carried out on two different groups of dogs; rewrite.

Lines 252-256: As I mentioned above, the reference values ​​must go in the Methodology section.

Author Response

We thank you for your constructive analysis of our manuscript, following your annotations and suggestions we have improved quality and contents of our work. Thank you for your notes. In attachment I send you my responses for your comments.  

This manuscript is a resubmission of an earlier submission. The following is a list of the peer review reports and author responses from that submission.

Round 1

Reviewer 1 Report

The authors present a study on the effect of exercise on urine albumin in dogs, as transient albuminuria has been reported in humans after exercise but information is lacking on this in dogs.  Many veterinarians extrapolate from the human evidence and recommend restriction of exercise before urinalysis for this reason.

Overall strengths:  The information is relevant and important, providing potentially significant information on canine urinary tract physiology.  The writing in English is good with minor grammatical edits suggested throughout the text. 

Overall weaknesses:  The study focuses on urine albumin, yet this value is left off of the table of results (Table 2) and not described in the text other than "Urine albumin was significantly higher for group of all dogs and female dogs after exercise than before."  The results reported on lines 171-172 that the urine albumin was significantly increased directly contradicts the author's conclusion that "30-40 minutes of physical effort, even in hot weather, should not lead to albuminuria," as they report in the results that it was "significantly increased."  Was this a typographical error?  Should it read "insignificantly increased?"  If so, suggest that the authors re-state it as "There was no significantly increase in urine albumin" to help avoid confusion.We need to see the actual numerical values, which were not included in the results table, to see for sure.

In addition, serum albumin, which could affect urine albumin, was not reported as having been evaluated in this study. While not having the values for serum albumin in comparison to urine albumin does not change the actual results of urine albumin change post-exercise, it makes it difficult to interpret why this result may be occurring.  The authors should list this as a limitation of their study and discuss what impact this may have had on the results.

1. The title "impact of high temperature on post-exercise albuminuria in dogs" may not be appropriate as 25 C (77 F) is not considered a high temperature in many parts of the world where interested readers may work.  For example, 25 C would be considered "mild" temperature in the Southern USA.  Similarly, the dogs' body temperatures did not reach clinically high values for exercising dogs (39.68 all dogs, 39.74 male dogs).   However, I realize that response to environmental temperature in dogs is based on acclimation, and in this population of dogs, 25 C might be considered a high temperature for them to experience.  Since temperature "feeling" (hot, mild, etc) is relative to acclimation, perhaps the authors could clarify within the materials and methods that this temperature is 'hot' based on typical temperatures of the location of the study.  

2. Line 35: 'this means that there is no need to limit the movement..."  This terminology seem to definitive to use after one study in 13 dogs.  Please consider changing to "This suggests that..." rather than "This means that..."  

3. Line 40-41: Urinalysis was the first laboratory test performed in medicine..."  This is an interesting fact and relevant to include here (I remember learning this in school) but would be good to support with a citation.

4. Lines 52-53; Studies conducted on dogs indicate that it might be expected to develop microalbuminuria after exercise is between 0 and 15%- while in 15% positive microalbuminuria was present both before and after exercise."  There is a problem with grammar in this sentence that makes the statement unclear.  I believe the authors are saying that in previous studies, 0 to 15% of the dogs had microalbuminuria, but cannot be sure.  Can the authors clarify this?

5. Line 54: "In contrast to human..."  grammatical suggestion: this is a sentence fragment.  Suggest "This is in contrast to human medicine where..."

6. Lines 60-61: "effective renal flow till..." and "Till now a lot of veterinarian...' (Grammar/slang) Suggest "effective renal flow until..." and 'currently, veterinarians...

7. Lines 69-70:  "This study included 13 healthy dogs 8 females (all after ovariohysterectomy) aand 5 males (after castration)..."  This line reads awkwardly in English. Suggest "this study included 8 ovariohysterectomized female dogs, 3 castrated dogs, and 5 intact male dogs."

8. Lines 75: In English, dog breeds should be capitalized:  For example: nova scotia duck tolling retrievers" should read "Nova Scotia Duck Tolling Retrievers."

9. line 99: During the 8 hours fasting period or at any time during the study, was water provided to the dogs?  If so, please note that in the study protocol section and discuss what impact that might have had on results.  If it was not offered during this time period, please note that in the study protocol description.

10. Line 103, the last word of the first sentence: "...(HGB), Hct, RBC, WBC were measure." Should read "(HGB), Hct, RBC, and WBD were measured."

11. Line 104: "...plan tubes."  should this be "plain" tunes, meaning without additives?

12. Line 112 regarding urine collection times: Were the dogs allowed access to water during or after exercise? If so, can the authors describe this and how it may or may not have affected UA results?

13. line 141:  "Exercise has been started..."  Grammar suggestions:  "Exercise was started..."  

14. Line 151: "The trainings was conducted..."  Grammar suggestion "The training was conducted..."

15. Line 165: Table 2 formatting issue:  The "All Dogs" values should be moved down one row for each value reported.

16. Line 221:  "...non-enzymatic dehydration" looks like a typographical error.  Should this be "non-enzymatic degradation...?"  

17. 222-223: "Males have a higher percentage of muscle in their body weight than females which have a higher percentage of body fat."   The citation used (21) does not support this statement. The study cited only studied three small breed male dogs before and after neutering and did not compare them to females dogs.  If this statement it citing something from the study's introduction or discussion section, recommend that the authors cite the original study instead.  Regarding body fat, this could only be compared in male and female dogs of similar size, breed, age, etc.  A well conditioned female dog an have less body fat percentage than an unconditioned, obese male dog, so advise the authors agains generalizing this statement without clarifying this specifically.

18. Line 227: "Urine took during the day..." Grammatical issue (slang).  Suggest saying "Urine obtained during the day..."

19: Line 246; "It also turns out that male are more likely to overhead than females, because they have bigger muscle mass."  The current evidence on canine heat injury does not support males being more at risk for overheating or heat-related injury than female dogs.  Multiple studies have shown no difference in risk based on sex.  Also, the term "it turns out that" is informal slang that suggests the authors studied that situation and they are commenting on this result.  Suggest the authors omit that from statements made in their manuscript if not directly discussing their results.  

Reviewer 2 Report

The authors show that dogs do not show pathological values ​​of albuminuria after exercising in a medium-high temperature environment.

The assessment of albuminuria in dogs after exercise has been previously evaluated in other studies, as cited by the authors. However, the study requires substantial modifications. The main limitation, as the authors confirm, is the low number of dogs studied. Furthermore, the results are based on a single urine sample before and after exercise, which increases the limitation of the results.

The methodology has some shortcomings: for example, they do not explain the methodology used to measure albumin and creatinine in urine.

English language requires substantial modifications, there are parts of the text that are difficult to understand.

The authors conclude that the animals subjected to physical effort do not show more albuminuria, however the results show statistically significant differences in the albuminuria and UAC values ​​before and after exercise, which is not discussed later.

Some references (?) are indicated as superscripts, I do not know their meaning or, if they are references, I have not seen the bibliography in the manuscript. Therefore, I have not been able to verify some of the statements in the manuscript.

The discussion needs substantial improvements and greater bibliographic relevance in the authors' claims.

Other details:

The bibliographic citation [9] (line 136) does not correspond to the one provided in the reference list

Line 143: How was the speed at which dogs trained measured?

Lines 237-238: If these are new results, they should be properly indicated in the results section. It is not correct to present new results in the discussion section.

Table 1 requires adding the units of the evaluated values ​​as well as the reference values ​​of the machines used.

Table 2 is missing the albuminuria data.